# Neurotransmitter recognition by human vesicular monoamine transporter 2

Dohyun Im [1,12] ✉, Mika Jormakka[1,12], Narinobu Juge[2,3,12], Jun-ichi Kishikawa [4,5], Takayuki Kato [5], Yukihiko Sugita [6,7,8], Takeshi Noda[6,7,9], Tomoko Uemura[1], Yuki Shiimura[1,10], Takaaki Miyaji [2,3], Hidetsugu Asada[1] & So Iwata[1,11] ✉

Human vesicular monoamine transporter 2 (VMAT2), a member of the SLC18 family, plays a crucial role in regulating neurotransmitters in the brain by facilitating their uptake and storage within vesicles, preparing them for exocytotic release. Because of its central role in neurotransmitter signalling and neuroprotection, VMAT2 is a target for neurodegenerative diseases and movement disorders, with its inhibitor being used as therapeutics. Despite the importance of VMAT2 in pharmacophysiology, the molecular basis of VMAT2-mediated neurotransmitter transport and its inhibition remains unclear. Here we show the cryo-electron microscopy structure of VMAT2 in the substrate-free state, in complex with the neurotransmitter dopamine, and in complex with the inhibitor tetrabenazine. In addition to these structural determinations, monoamine uptake assays, mutational studies, and pKa value predictions were performed to characterize the dynamic changes in VMAT2 structure. These results provide a structural basis for understanding VMAT2-mediated vesicular transport of neurotransmitters and a platform for modulation of current inhibitor design.

Monoamine neurotransmitters, including dopamine, serotonin (5-HT; 5-hydroxytryptamine), histamine, and noradrenaline, are biochemical messengers that play a pivotal role in the central nervous system. These neurotransmitters exert regulatory influence over neural functions encompassing aspects such as emotion, learning, motor control, sleep, and wakefulness. Disruptions in the levels of these neurotransmitters within the brain have been linked to various psychiatric and neurodegenerative disorders. Consequently, maintaining the precise control of monoamine neurotransmitters in the brain is of paramount importance.

Vesicular monoamine transporters (VMATs) are membrane transporters responsible for the transport of monoamines. They are categorized into two distinct subtypes: VMAT1 and VMAT2[1,2]. VMAT1 is primarily expressed in the peripheral nervous system and is responsible for transporting neurotransmitters within vesicles in the gastrointestinal tract, kidney, adrenal glands, and parasympathetic nervous system. In contrast, VMAT2 is predominantly found in the central nervous system and facilitates the transport of monoamines within synaptic vesicles in the neuron. Alongside the vesicular acetylcholine transporter (VAChT) and the vesicular polyamine transporter (VPAT), VMAT1 and VMAT2 are classified within the SLC18 family, which is a part of the Solute Carrier (SLC) transporter superfamily (Supplementary Fig. 1)[3–5]. VMAT2 is furthermore a member of the Major facilitator superfamily (MFS), the largest superfamily of secondary active transporters[6,7].

[1]Department of Cell Biology, Graduate School of Medicine, Kyoto University, Kyoto, Japan. [2]Department of Genomics and Proteomics, Advanced Science Research Center, Okayama University, Okayama, Japan. [3]Department of Molecular Membrane Biology, Graduate School of Medicine, Dentistry and Pharmaceutical Sciences, Okayama University, Okayama, Japan. [4]Department of Applied Biology, Kyoto Institute of Technology, Kyoto, Japan. [5]Institute for Protein Research, Osaka University, Suita, Japan. [6]Laboratory of Ultrastructural Virology, Institute for Life and Medical Sciences, Kyoto University, Kyoto, Japan. [7]Laboratory of Ultrastructural Virology, Graduate School of Biostudies, Kyoto University, Kyoto, Japan. [8]Hakubi Center for Advanced Research, Kyoto University, Kyoto, Japan. [9]CREST, Japan Science and Technology Agency, Kawaguchi, Japan. [10]Institute of Life Science, Kurume University, Kurume, Japan. [11]RIKEN SPring-8 Center, Sayo-gun, Japan. [12]These authors contributed equally: Dohyun Im, Mika Jormakka, Narinobu Juge. ✉e-mail: im.dohyun.3s@kyoto-u.ac.jp; iwata.so.2z@kyoto-u.ac.jp

VMAT2 is responsible for the accumulation of monoamines within synaptic vesicles at axon terminals in exchange for protons using an antiport mechanism (Fig. 1a)[8]. It regulates the uptake of these substances from the extracellular synaptic cleft into the vesicle, facilitating their recycling within the synaptic vesicle. Consequently, VMAT2 represents a crucial pharmaceutical target in the therapeutic intervention of psychiatric and neurodegenerative disorders[9]. As such, inhibitors targeting VMAT2 induce monoamine depletion within neurons and have been employed in the treatment of neurodegenerative diseases like Huntington's disease[10] and tardive dyskinesia induced by antipsychotic medications[11,12]. In addition, reserpine, an irreversible VMAT2 inhibitor, is widely used to treat hypertension[13].

Previous studies based on homology models and mutagenesis experiments have proposed the location of the substrate-binding site and transport mechanisms for VMAT2[14–16]. In addition, several recent studies have reported the structure of VMAT2[17,18]. However, due to the limited structural and functional information, comprehensive molecular details surrounding substrate recognition and its transport mechanism have yet to be detailed.

In this study, we report the cryogenic electron microscopy (cryo-EM) structures of the human VMAT2 in the apo state, bound to the substrate dopamine, and bound to the inhibitor tetrabenazine. These results provide important molecular details of VMAT2 substrate and inhibitor binding. Based on this structural information, we performed the monoamine uptake analysis through mutagenesis introduction, prediction of the pKa values of key polar residues, and identified specific residues that specifically intervene in the dynamic changes of VMAT2 structure. Furthermore, our results allow us to construct a working model for proton-coupled monoamine neurotransmitter transport. As such, the results will enhance our understanding of neuronal molecular mechanisms, as well as facilitate further modification of previously developed drugs and guide the development of new therapeutics.

## Results

### Cryo-EM structure of VMAT2

To generate a VMAT2 protein conducible for structural studies, we systematically screened various constructs for optimal expression and stability, ultimately selecting a construct lacking the C-terminus and a portion of the extracellular loop (ECL) 1 (Supplementary Fig. 2). Due to its relatively small size, we generated a monoclonal antibody raised against VMAT2 as a fiducial marker to aid in single-particle analysis. By utilizing the Fab fragment of this antibody, we successfully determined the structures of human VMAT2 in the apo state (VMAT2$_{apo}$), in complex with dopamine (VMAT2$_{dop}$) and in complex with the inhibitor tetrabenazine (VMAT2$_{tet}$) at resolutions of 3.05 Å, 2.90 Å and 3.18 Å, respectively (Fig. 1b, Table 1, and Supplementary Figs. 3, 4). The Fab portion binds to the intracellular region of VMAT2 and does not notably affect the transport activity of VMAT2 (Fig. 1c).

The overall structure is divided into two domains, an N-terminal domain comprising transmembrane helices (TM) 1–6 and a C-terminal domain comprising TM7-12, an architecture typical for the MFS transporter fold[19]. In the general transport mechanism of alternating access in MFS transporters, the N-terminal and C-terminal bundles reorganize symmetrically around the central cavity (substrate-binding site), allowing for sequential access to the opposite sides of the membrane[7]. The structure of VMAT2$_{apo}$ adopted a conformation with the central cavity orienting towards the lumen of the vesicle and represents the outward-facing conformation (Fig. 1d). The bottom of the central cavity is closed off by the intracellular ends of TMs 4 and 5 in the N-terminal domain and TMs 10 and 11 on the C-terminal domain, forming an "intracellular gate" stabilized primarily by hydrophobic interactions (Fig. 1d). In addition, the triad of the guanidino group of Arg217 (TM5), the carboxylate group of Asp411 (TM10), and the hydroxyl group of Tyr418 (TM11) forms a stabilizing hydrogen-bonded network (Supplementary Fig. 5). Previous studies of rat VMAT2 indicated that Arg218 and Tyr419 (corresponding to Arg217 and Tyr418 in humans, as shown in Supplementary Fig. 1) interacts and play a part in

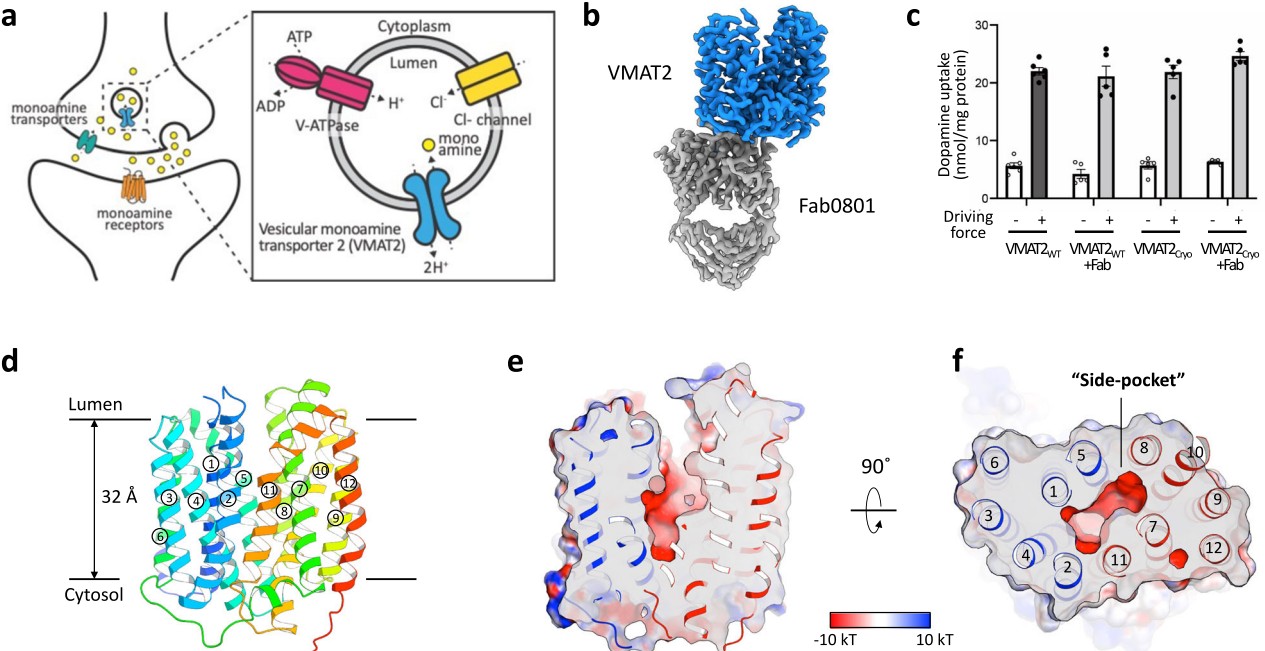

**Fig. 1 | Overall structures of the human VMAT2. a** Mechanism of monoamine uptake into vesicles by VMAT2. **b** Cryo-EM density of VMAT2$_{apo}$-Fab0801 complex at a threshold level of 1.2. **c** Dopamine transport activity of wild-type VMAT2 (VMAT2$_{WT}$) and cryo-EM construct (VMAT2$_{Cryo}$) in the presence or absence of the intracellular recognition antibody Fab0801. Proteoliposomes containing purified VMAT2 protein were prepared in buffer at pH 6.0, incubated in buffer at either pH 6.0 (−) or pH 7.5 (+) for 1 min. Error bars represent mean ± SEM; points represent biologically independent measurements (n = 6 for VMAT2$_{WT}$; n = 5 for others). **d** Cartoon representation of VMAT2$_{apo}$. **e, f** Electrostatic surface representation of VMAT2$_{apo}$ highlighting the central cavity. Side view (**e**) and top view (**f**).

## Table 1 | Cryo-EM data collection, refinement and validation statistics

| | VMAT2$_{apo}$ (EMDB-37774) (PDB 8WRD) | VMAT2$_{dop}$ (EMDB-37775) (PDB 8WRE) | VMAT2$_{tet}$ (EMDB-37867) (PDB 8WVG) |
|---|---|---|---|
| **Data collection and processing** | | | |
| Magnification | 81,000 | 81,000 | 81,000 |
| Voltage (kV) | 300 | 300 | 300 |
| Electron exposure (e–/Å$^2$) | 65 | 65 | 65 |
| Defocus range (µm) | −0.8 to –1.8 | −0.8 to –1.8 | −0.8 to –1.8 |
| Dose rate (e–/Å$^2$/s) | 7.499 | 7.326 | 7.108 |
| Pixel size (Å) | 0.88 | 0.88 | 0.88 |
| Symmetry imposed | C1 | C1 | C1 |
| Initial particle images (no.) | 23,399,580 | 7,536,420 | 9,417,729 |
| Final particle images (no.) | 1,286,931 | 1,297,524 | 697,706 |
| Map resolution (Å) | 3.05 | 2.90 | 3.18 |
| FSC threshold | 0.143 | 0.143 | 0.143 |
| **Refinement** | | | |
| Initial model used (PDB code) | AF2 model | 8WRD | 8WRE |
| Model resolution (Å) | 3.2 | 3.1 | 3.3 |
| FSC threshold | 0.5 | 0.5 | 0.5 |
| Map sharpening B factor (Å$^2$) | −81.42 | −52.86 | −120.46 |
| Model composition | | | |
| Non-hydrogen atoms | 6327 | 6329 | 6361 |
| Protein residues | 830 | 829 | 831 |
| Ligands | – | LDP: 1 | XEQ: 1 |
| B factors (Å$^2$) | | | |
| Protein | 50.68 | 50.60 | 51.19 |
| Ligand | | 20.00 | 28.15 |
| R.m.s. deviations | | | |
| Bond lengths (Å) | 0.003 | 0.003 | 0.004 |
| Bond angles (°) | 0.551 | 0.537 | 0.610 |
| Validation | | | |
| MolProbity score | 1.42 | 1.20 | 1.58 |
| Clashscore | 4.66 | 3.71 | 4.88 |
| Poor rotamers (%) | 0.14 | 0.14 | 0.00 |
| Ramachandran plot | | | |
| Favored (%) | 96.96 | 97.81 | 95.38 |
| Allowed (%) | 3.04 | 2.19 | 4.62 |
| Disallowed (%) | 0 | 0 | 0 |

the intracellular gate, and mutagenesis of these residues abrogated transport activity[15].

The central cavity exhibits polarity and is distinctive in that there is a "side-pocket" on the C-terminal domain side of the cavity, surrounded by TM7, 8, 10, and 11 (Fig. 1e, f). This "side-pocket" is similar to what has been observed in other MFS-type transporters, such as GLUTs and ferroportin, and is commonly a site for substrate binding[20,21].

VMAT2 possesses a long loop, ECL1 (Pro42–Val129), located between TM1 and TM2 that faces the intravesicular space. Cys117 within this loop is believed to contribute to structural stabilization by forming a disulfide bond with Cys324 in the C-terminal bundle. Some MFS transporters share a similar topology, and ECL1 acts as a lid to cover the extracellular side have been reported (e.g., STP10, PCFT)[22,23]. However, in the VMAT2 construct used in the cryo-EM study, this specific loop was deleted in our construct screening. Nevertheless, the loop-deficient protein exhibited transport activity equivalent to that of

the wild-type of VMAT2 (Fig. 1c and Supplementary Fig. 6c). This loop does not seem to be important for VMAT2 transport activity, although it may be involved in other functional aspects, such as regulation.

### Dopamine recognition in VMAT2

To gain a comprehensive understanding of the role of monoamines at synaptic terminals, it is imperative to elucidate the mechanisms underlying the uptake of monoamines into vesicles and the recognition of substrate. Therefore, in addition to the apo structure, we determined the structure of an outward-facing conformation of VMAT2 bound to dopamine. The structure of VMAT2$_{dop}$ shows that the dopamine-binding site is located in the "side-pocket" on the C-terminal bundle side, with residues from TM5, 7, 8, 10, and 11 participating in the binding (Fig. 2a, b). The overall location of the dopamine-binding site is reminiscent of what was observed in GLUT3, a sugar transporter[20]. The substrate-binding site in GLUT3, as in the case of dopamine in our VMAT2$_{dop}$ structure, was situated in a "side-pocket" on the C-terminal domain side, although dopamine is located closer to TM8 (Fig. 2c). In VMAT2$_{dop}$, the two catechol hydroxyl groups of dopamine forms hydrogen bonds with Glu312 of TM7 and Ser338 of TM8, while the primary amine of dopamine forms a salt bridge with Asp399 on TM10 and a stabilizing hydrogen bond with Asn305 of TM7 (Fig. 2d, e). This binding mode is reminiscent of what was observed in a substrate-bound dopamine transporter (Supplementary Fig. 7)[24]. Salt bridges between amine ligands and negatively charged residues are also highly conserved in aminergic G protein-coupled receptors (GPCRs) (Supplementary Fig. 7)[25–28].

Furthermore, the amine group of dopamine interacts with Tyr341 of TM8 and Tyr433 of TM11, forming a sandwich-like arrangement. Tyr341 also contributed to π−π stacking interactions with the benzene ring of catechol and appears to play a significant role in dopamine binding (Fig. 2d, e).

The importance of the residues coordinating dopamine in our structure was confirmed by mutagenesis and transport activity measurements in a reconstituted transport assay (Fig. 2f and Supplementary Fig. 6d). In this assay even the "mild" mutants N305Q and E312D illustrated an abolished transport activity. Previous studies of rat VMAT2 has pointed to a critical role of Glu312 in substrate/inhibitor binding as mutations to this residue rendered protein unable to bind [$^3$H]-dihydrotetrabenazine ([$^3$H]-TBZOH), an analog of the non-competitive inhibitor tetrabenazine[29]. In addition, we observed a reduced transport rate of a D399N mutant, while a D399A mutation rendered the protein inactive (Fig. 2f and Supplementary Fig. 6d)[29].

During the preparation of our manuscript, the a structure of VMAT2 was reported[17]. The report detailed the structure of VMAT2 in complex with serotonin (5-HT; VMAT2$_{5HT}$), another endogenous substrate of VMAT2, in an inward-facing conformation. Comparing the published 5-HT bound structure with our VMAT2$_{dop}$ structure illustrates that they exhibit different substrate-binding modes (Supplementary Fig. 8). Although the conformation of the ring architecture of both ligands was relatively similar, the orientation of the ethyl chain is completely different (Supplementary Fig. 8b). The amine portion of dopamine is pointing towards the intracellular side, whereas the ethyl chain of serotonin is parallel to the membrane plane, aimed towards the center of the transporter. The orientational difference of the ethyl chain leads to drastic differences in substrate coordination with the primary amine of serotonin, forming a salt bridge with Glu312. These different substrate-binding modes can potentially be attributed to the outward vs inward conformations of VMAT2$_{dop}$ and VMAT2$_{5HT}$ or reflect multiple substrate-binding modes in VMAT2 (Supplementary Fig. 8a).

### Tetrabenazine-binding site

Tetrabenazine is the first drug approved in the U.S. for the treatment of Huntington's disease and is used for hyperactive movement

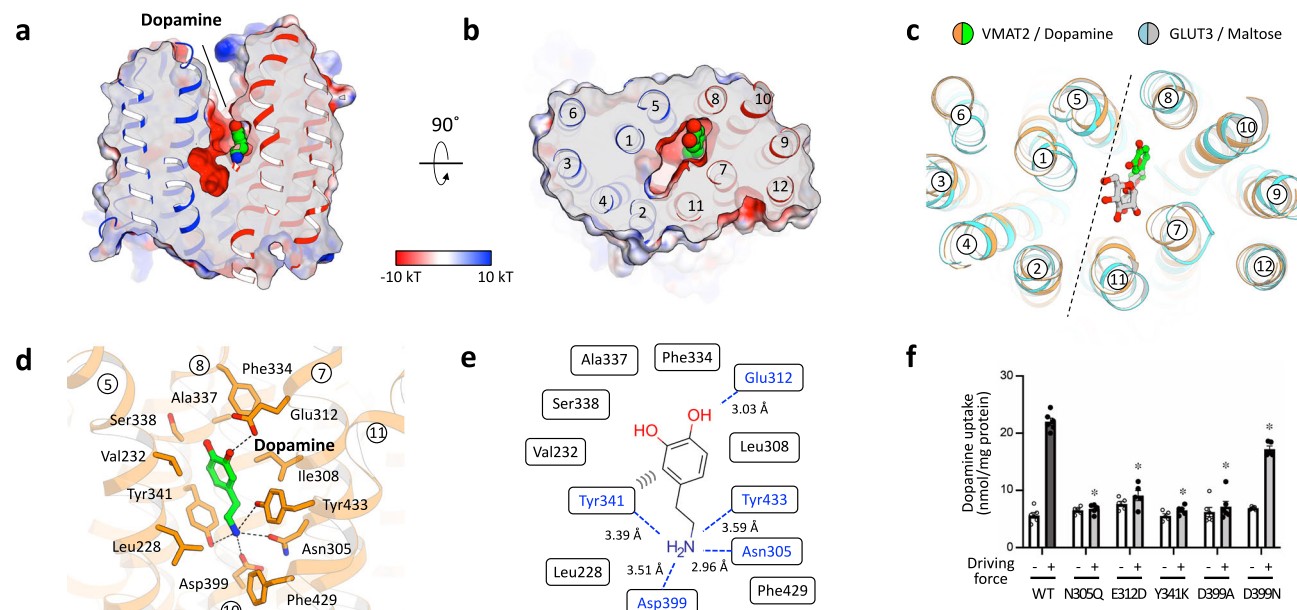

**Fig. 2 | Dopamine recognition of VMAT2. a, b** Electrostatic surface representation of VMAT2$_{dop}$ highlighting the substrate-binding pocket. Side view (**a**) and top view (**b**). **c** Top view of the structural superposition of VMAT2$_{dop}$ (orange) and the outward-facing conformation of GLUT3 (cyan, 4ZWC). Dopamine of VMAT2 and maltose of GLUT3 are indicated by green and gray sticks. **d** Detailed interaction between dopamine and VMAT2. Dopamine and contact residues are shown as green and orange sticks. **e** Schematic representation of dopamine-binding interaction. Hydrogen bonds and salt bridge are shown as dashed lines. **f** Uptake of dopamine by proteoliposomes containing purified VMAT2 variants. Error bars represent mean ± SEM; points represent biologically independent measurements ($n = 6$ for WT and D399A; $n = 5$ for others). $P$ values for differences between wild-type and variants were obtained from a one-way analysis of variance (ANOVA) followed by the Dunnett's multiple comparisons test. *$P < 0.01$; from left to right, $P < 0.0001$, $P < 0.0001$, $P < 0.0001$, $P < 0.0001$, and $P = 0.0003$.

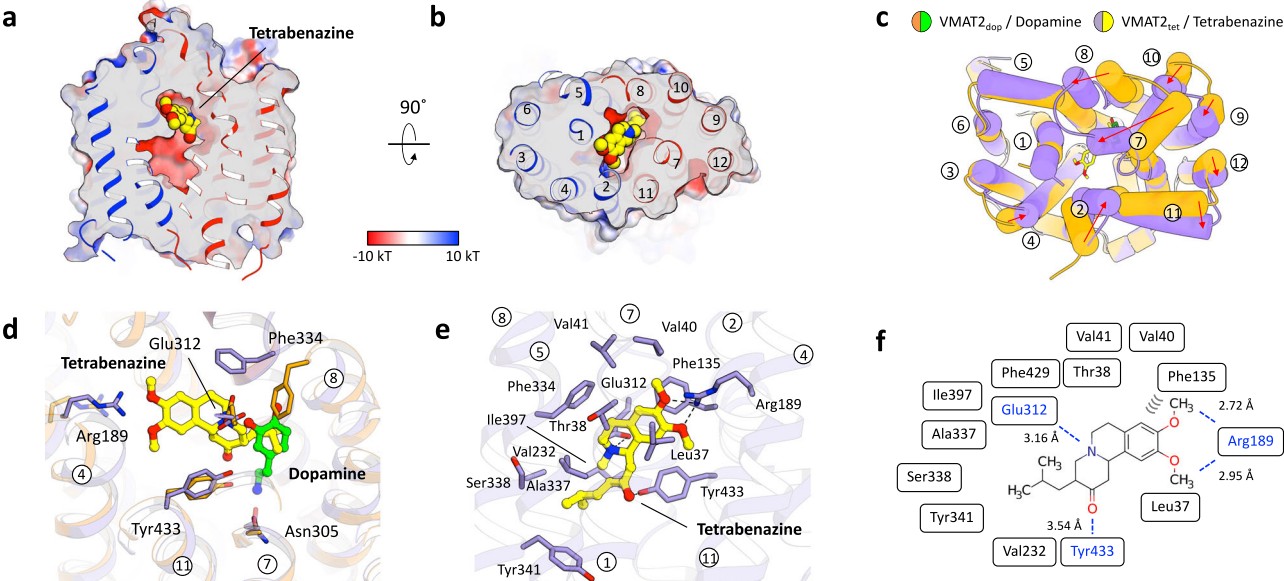

**Fig. 3 | The tetrabenazine-binding site of VMAT2. a, b** Electrostatic surface representation of VMAT2$_{tet}$ highlighting in the inhibitor binding pocket. Side view (**a**) and top view (**b**). **c, d** The structural comparison between VMAT2$_{tet}$ (purple) and VMAT2$_{dop}$ (orange). Tetrabenazine and dopamine are indicated by yellow and green sticks. The helices are shown as cylinders. Top view (**c**) and close-up view (**d**). Residues that do not act on the other ligand are shown as transparent (**d**). **e** Detailed interaction between tetrabenazine and VMAT2. The inhibitor and contact residues are shown as yellow and purple sticks. **f** Schematic representation of tetrabenazine-binding interaction. Hydrogen bonds and salt bridge are shown as dashed lines.

disorder[30]. However, the molecular mechanism of tetrabenazine action in VMAT2 is still unknown. To gain insight into tetrabenazine binding, we determined a VMAT2$_{tet}$ structure at 3.18 Å where we can observe a strong and distinct density peak in the center of the central cavity, representing tetrabenazine (Fig. 3a, b and Supplementary Figs. 4c, f). Compared with the VMAT2$_{dop}$ structure, the lower half of

the transporter (intracellular gate) remains almost identical, while the upper half (lumen side) shows significant conformational changes (RMSD: 0.732 Å). These structural changes, particularly in TMs 2, 4, 7, 8, and 10, which have shifted toward the central cavity, lead to the complete closure of luminal access (Fig. 3c). As such, VMAT2$_{tet}$ adopts an occluded conformation (Fig. 3a–c).

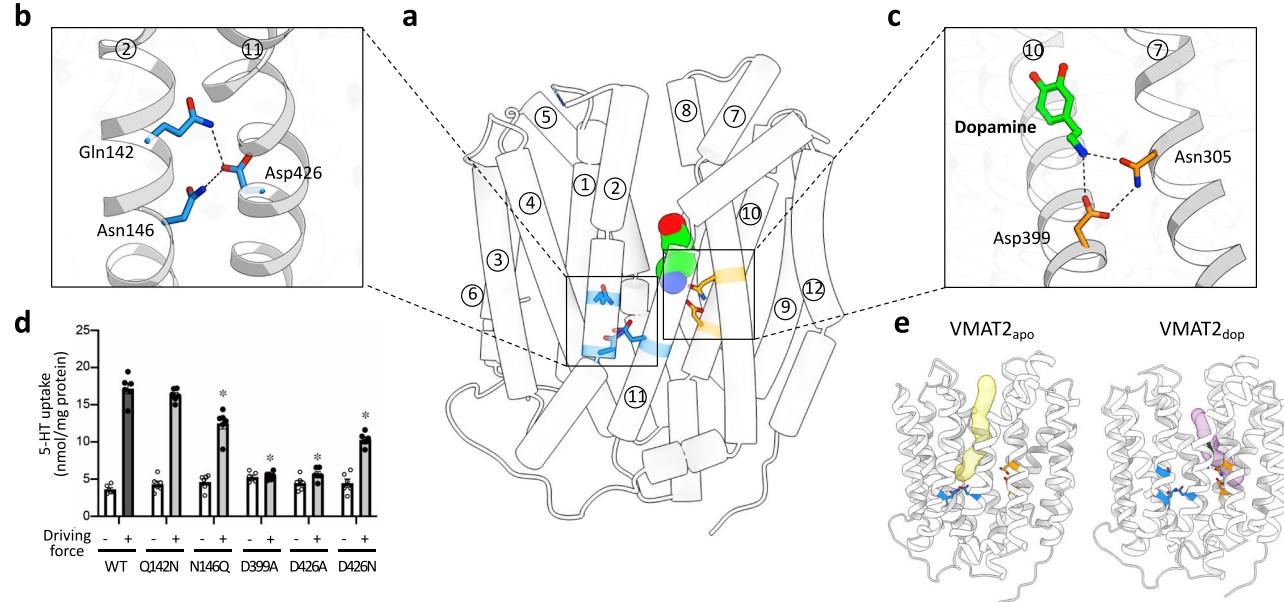

**Fig. 4 | Two functional interaction networks for potential proton-coupling.**
**a**–**c** Putative proton-coupling sites of the VMAT2$_{dop}$. Side view (**a**) and close-up of the hydrogen-bonded network (**b**) and substrate-binding site (**c**). The critical residues are represented by blue and orange sticks, and their interactions are indicated by dashed lines. **d** Uptake of 5-HT by proteoliposomes containing purified VMAT2 variants. Error bars represent mean ± SEM; points represent biologically independent measurements ($n = 6$). $P$ values for differences between wild-type and variants were obtained from a one-way ANOVA followed by the Dunnett's test. *$P < 0.01$; from left to right, $P = 0.0011$, $P < 0.0001$, $P < 0.0001$, and $P < 0.0001$. **e** The lumen-accessible tunnels in VMAT2$_{apo}$ (left) and VMAT2$_{dop}$ (right).

Whereas dopamine predominantly utilizes residues on the C-terminal domain for binding, tetrabenazine actively interacts with residues on both of the N-terminal and C-terminal domain by occupying the entire central cavity (Fig. 3c, d). A broad range of residues in TM1, 2, 4, 5, 6, 7, and 11 are involved in tetrabenazine binding (Fig. 3e). Tetrabenazine consists of a tricyclic skeleton with a methylpropyl in the tail and a tentacle-like dimethoxy group on the opposite side (Fig. 3f). The amine in the tricyclic portion forms an ion pair with Glu312 of TM5, and the ring structure is entirely covered and stabilized by hydrophobic residues, including Leu37, Phe135, Val232, Phe334, Phe429, and Tyr433 (Fig. 3e, f). The coordination of the methylpropyl moiety is in close proximity to the site of the benzene ring in dopamine (Fig. 3d). This region forms an alkyl interaction with Val232 and Ala337. The dimethoxy group on the opposite side also makes interactions, in particular Arg189 in TM4 which is hydrogen-bonded to both methoxy groups (Fig. 3e, f). It is also stabilized by Val40 and Val41 of TM1 through alkyl interactions. During the preparation of this manuscript, Pidathala et al. also published a tetrabenazine-bound VMAT2 structure (Supplementary Fig. 8c, d)[17]. This structure is virtually identical to our structure, thus both studies agree well on the binding mode of tetrabenazine.

### Putative proton-coupling sites

VMAT2 is recognized for its ability to transport monoamines into vesicles by performing countertransport (antiport) of H$^+$ and monoamines across the vesicular membrane in a 2H$^+$/monoamine stoichiometry (Fig. 1a)[8]. To gain a detailed understanding of the transport mechanism of VMAT2 it is important to identify residues involved in the proton-coupling, which often involves acidic residues (Asp and Glu) responsible for sensing [H$^+$]. Therefore, we examined the ion pair interactions observed in the structures obtained in this study and identified several residues of interest (Fig. 4a–c). First, we focused our attention on residues found deep within the central cavity, a region which connects the N-terminal bundle to the C-terminal bundle (Fig. 4b). Within this area, Asp426 of TM11 interacts with Gln142 and Asn146 of TM2, contributing to the stabilization of the outward-facing

conformation. To evaluate the importance of these residues, we generated mutant proteins and evaluated the transport activity in proteoliposomes. Indeed, we found that when Asp426 was changed to either an Asn or Ala residue, there was a marked reduction in substrate transport activity. Furthermore, mutating the interacting residue Asn146 to glutamine (N146Q) also displayed decreased transport activity (Fig. 4d). This suggests that this interaction is crucial for maintained protein function. However, a "mild" mutation of Gln142, Q142N, had no significant change in transport activity in our assays. In contrast, in a previous study mutagenesis of the equivalent residue in rat VMAT2 to alanine (Q142A) illustrated a significant reduction in 5-HT transport[14].

In the structure of VMAT2$_{apo}$, Asp426 is facing a lumen-accessible solvent channel, which may facilitate protonation (Fig. 4e). To assess this further, we used several pKa prediction servers to estimate the protonation/deprotonation state of VMAT2 residues based on the structures[31,32]. The resulting predictions indicate that Asp426 has a relatively high pKa value (pKa -6–9) compared to other acidic residues (Supplementary Fig. 9). Collectively, these observations suggests that Asp426 is a potential proton-coupling site in VMAT2. In the structure of VMAT2$_{dop}$ a second solvent-accessible channel can also be observed, which is leading from the luminal side to the dopamine-binding site (Fig. 4e). At the end of this channel lies Asp399 which, as described above, is facing the substrate dopamine and recognizes its amine portion while also forming an ion pair with Asn305 of TM7 (Fig. 4c). In pKa predictions, Asp399 exhibited a pKa value similar to that of Asp426, if not slightly higher (Supplementary Fig. 9). The interaction between Asp399 and Asn305 is likely disrupted in the D399A, D399N, and N305Q mutants, all of which have significantly reduced transport activity (Fig. 4d). We also estimated the pKa of Asp399 and Asp426 of the recently determined inward-facing conformation of VMAT2[17] using the same prediction servers. While the difference in pKa was relatively modest when compared to the outward-facing conformation (VMAT2$_{dop}$), it is important to take into consideration the local environments; the intracellular pH of neurons is 7.1–7.5, while the lumen of neural vesicles have an acidic

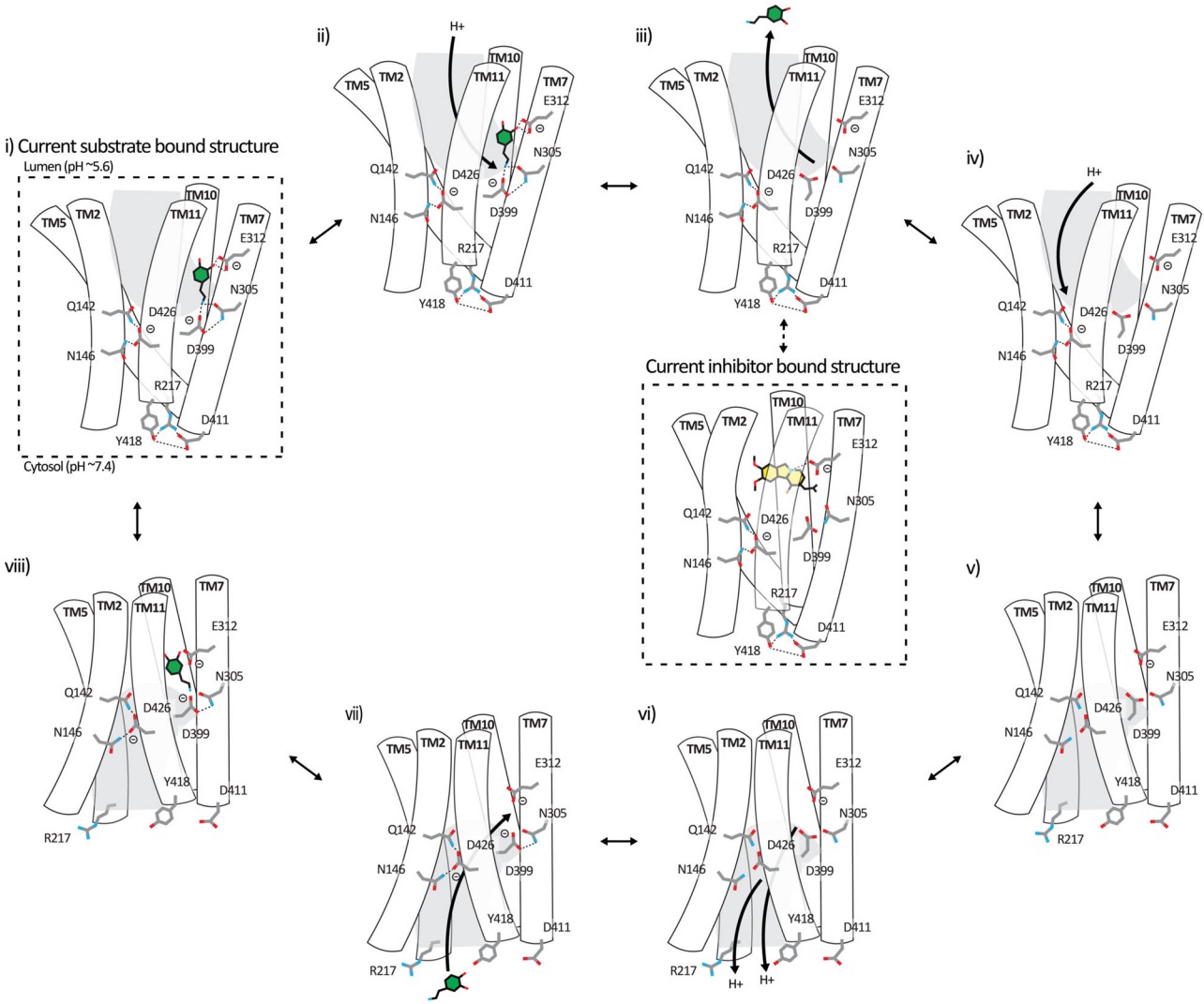

**Fig. 5 | Proposed alternating-access transport mechanism. i–viii** Diagram illustrating the conformational states explored by VMAT2 during its transport cycle. The dashed square directly arises from our data, while the other panels are conjectured based on our findings and prior insights into related transporters.

intravesicular pH 5.6[33,34]. The equivalent residue to Asp399 in rat VMAT1 was subject to previous mutagenesis studies, where mutation to either Ser or Cys abolished both transport and binding of 5-HT[35]. In contrast, when replaced with a glutamic acid the binding of substrate and transport activity was restored, but with a shifted pH optimum for transport. Similarly, studies of rat VMAT2 showed that an Asp or Glu is essential at this position for transport activity[14]. Collectively, these observations suggest a dual role of Asp399, in which it serves a critical role in substrate binding while also being a proton-coupling site.

## Discussion

VMAT2 mediates the accumulation of neurotransmitters in synaptic vesicles by utilizing a proton gradient across the vesicular membrane. We sought to determine the structure of VMAT2 to gain molecular insights into the transport mode and to detail the substrate-binding site. As such, we determined the cryo-EM structures of VMAT2 in the apo, dopamine-bound and tetrabenazine-bound states, which together with functional studies provide a framework for understanding the transport mechanism of VMAT2, as well as its inhibition. Furthermore, taken together our data enables us to propose a model for proton-coupled dopamine transport by VMAT2 (Fig. 5). In the first step (Fig. 5i), the substrate-bound transporter adopts an outward (luminal) facing conformation. The "intracellular gate" is closed and both

Asp399 and Asp426 are in their deprotonated state (Supplementary Figs. 10a). In this state, Asp399 forms a hydrogen bonding interaction with Asn305 and is also involved in dopamine coordination while Asp426 forms hydrogen bond cluster with the pair of Gln142/Asn146. This state is represented by our VMAT$_{dop}$ structure, which was obtained at pH 7.5 and putatively captures a state just prior to the substrate is released into the vesicle (Figs. 5ii–iv). The lumen in synaptic vesicle has a pH of ~5.6, while the predicted pKa of the Asp399/Asp426 residues in the outward-facing conformation indicates that this pair have an unusually high pKa (pKa > 6). Hence, when exposed to the luminal space in its physiological setting, these two residues are likely rapidly protonated. Based on our pKa predictions, we propose a hierarchical protonation of VMAT2 to ensure no substrate "backflow" into the cytoplasm. As such, we propose that Asp399 is protonated first, leading to disruption of the Asp399, Asn305 and substrate interaction, triggering the release of bound substrate. Following this, the protonation of Asp426 disrupts its interactions with the Gln142/Asn146 pair, weakening the stability of the outward-facing conformation. This latter protonation is potentially crucial for the transition into the inward-facing conformation through a cascade effect. Firstly, it causes a direct conformational change in TM11 as the stabilizing interaction between TM11 (Asp426) and TM2 (Gln142/Asn146) are disrupted. Secondly, the structural change in TM11

translates to long-range conformational changes at the "intracellular gate" causing disruption of the ionic and hydrogen bonding interactions in the Arg217 (TM5), Asp411 (TM10), and Tyr418 (TM11) triad. We propose that these changes are sufficient to trigger a transition from outward- to inward-facing conformation. As anecdotal support for this step, Pidathala et al.[17] "trapped" an inward-facing conformation of VMAT2 using a Y418S mutant protein (Supplementary Fig. 10b). It thus appears reasonable to assume that the formation of the triad is required for an efficient VMAT2 return to the outward-facing conformation. Conversely, disruption of this triad may be sufficient to drive the out-to-in conformational change. In previous studies of rat VMAT2, an interaction network involving Lys139–Gln143–Asp427 (corresponding to Lys138–Gln142–Asp426 in humans, as shown in Supplementary Fig. 1) was discussed as a potential hinge region between TM2 and TM11[33,36]. Our studies support the significance of this network and, based on our structural insights, have furthermore highlighted the importance of the Gln142–Asp426–Asn146 interaction network.

As the transporter assumes the inward-facing conformation (Fig. 5v–viii), the central cavity becomes exposed to the cytosol (pH ~7.4). In this conformation, we anticipated that the molecular environment has changed sufficiently for Asp399 and Asp426 (pKa < 6 in the inward-facing conformation; Supplementary Fig. 9) to become deprotonated. Since the structural information is limited, predicting events leading to a reversal of the conformational changes is challenging. However, the occluded structure indicates that there are significant structural changes occurring around the substrate-binding site during the transition to inward-facing conformation. As such, a reversal of this could be catalyzed through an induced fit by the substrate, facilitated by the deprotonated Asp399. Indeed, comparing the inward-facing reserpine structures (Pidathala et al. PDB 8T6B, and Wu et al. PDB 8JTC) to the serotonin-bound structure (Pidathala et al. PDB 8T6A) illustrates small but significant differences[17,18]. In particular, the intracellular halves of TMs 5, 8, and 10 have shifted towards the center of the protein (Supplementary Fig. 10c, d). As such, the serotonin-bound structure appears more "closed" than the reserpine structures, which we speculate is a step towards a closure of the intracellular gate and formation of an occluded state. This may be facilitated by deprotonation of Asp399/Asp426 or substrate-induced fit (or combination of the two). However, due to the Y418S mutation, the intracellular gate is unable to fully close.

While this constitutes a working model based on our results and observations, further studies are required to fully illuminate the detailed steps taking place. As such, we hope the results presented here will stimulate further studies into the transport mechanism of VMAT2. Furthermore, we hope that these structural insights will contribute to the development of treatments for psychiatric disorders and neurodegenerative diseases associated with VMAT2.

## Methods

### Constructs screening
The coding sequence of human VMAT2 (UniProt ID Q05940) was synthesized by Eurofins Genomics (Japan). VMAT2 was stabilized by removing the C-terminal 34 residues and the first extracellular loop (Ile51–Glu120) (Supplementary Fig. 2a). To obtain this stable construct, we performed high-throughput fluorescent-based screening in *Saccharomyces cerevisiae*[37–39]. Details of the construct screening were reported previously[25]. The construct was subcloned into the pFastBac1 vector (Invitrogen), with a C-terminus tobacco etch virus (TEV) protease cleavage site, green fluorescent protein (GFP), and an octa-histidine tag.

### Protein expression and purification
Recombinant baculoviruses were generated using the Bac-to-Bac baculovirus expression system (Invitrogen, Carlsbad, CA).

*Spodoptera frugiperda* (*Sf*9) insect cells were grown to a density of $1.5 \times 10^6$ cells/mL and infected with VMAT2 viral stocks at a multiplicity of infection (MOI) of 0.05 and were harvested 84 h later. Cell pellets were resuspended with hypotonic buffer (10 mM HEPES, pH 7.5, 20 mM KCl, and 10 mM MgCl$_2$) and were repeatedly washed and centrifuged in high osmotic buffer (10 mM HEPES, pH 7.5, 1 M NaCl, 20 mM KCl, and 10 mM MgCl$_2$) containing EDTA-free protease inhibitor cocktail (Nacalai Tesque) and 5 mM DTT to purify the cell membranes. The purified membranes were solubilized for 2 h at 4 °C in solubilization buffer (50 mM HEPES, pH 7.5, 500 mM NaCl, 1% (w/v) n-dodecyl-ß-D-maltopyranoside (DDM, Anatrace), 0.2% (w/v) cholesteryl hemisuccinate (CHS, Sigma-Aldrich), and 20% (v/v) glycerol) supplemented with 2 mg/ml iodoacetamide (Wako Pure Chemical Industries, Ltd.), 2.5 mM DTT, and a protease inhibitor cocktail tablet. Insoluble materials were removed by centrifugation, and the supernatant was incubated with TALON metal affinity resin (Clontech) for 10 h at 4 °C under slow stirring. The resin was subsequently loaded onto a gravity flow column (Econo-Column, Bio-Rad) and subsequently washed with wash buffer (20 mM HEPES, pH 7.5, 150 mM NaCl, 10% (v/v) glycerol, 0.05% (w/v) DDM, 0.01% (w/v) CHS, 10 mM imidazole, and 0.25 mM Tris (2-carboxyethyl) phosphine (TCEP)). The protein was eluted in elution buffer (20 mM HEPES, pH 7.5, 150 mM NaCl, 10% (v/v) glycerol, 0.018% (w/v) DDM, 0.0036% (w/v) CHS, 300 mM imidazole, and 0.25 mM TCEP) and concentrated to 2.5 ml with a 100-kDa molecular weight cutoff Amicon Ultra-15 concentrator (Millipore). The imidazole was removed using a PD-10 column (GE Healthcare). The desalted protein was incubated with His-tagged TEV protease (expressed and purified in-house) for 10 h. The TEV protease cleaved His-tagged GFP, and uncleaved protein were subsequently removed by passing the suspension through Ni Sepharose High-Performance resin (GE Healthcare). Finally, the protein was concentrated to 0.5 mL and subjected to size-exclusion chromatography on a Superdex 200 Increases 10/300 column (GE Healthcare) pre-equilibrated with a buffer containing 20 mM of HEPES (pH 7.5), 150 mM of NaCl, 0.018% (w/v) DDM, 0.0036% (w/v) CHS, and 2.5 mM TCEP. The fractions containing the monomeric proteins were collected for further experiments. For the VMAT2$_{tet}$ samples, lauryl maltose neopentyl glycol (LMNG; Anatrace) was used instead of DDM. DDM was replaced with LMNG during the wash process of TALON purification, and it was used in all subsequent processes.

### Antibody generation
All the animal experiments conformed to the guidelines of the Guide for the Care and Use of Laboratory Animals of Japan and were approved by the Kyoto University Animal Experimentation Committee (approval no. Med-kyo22055). As the antigen, we used a stabilized VMAT2. Purified antigen was reconstituted into liposomes containing chicken egg yolk phosphatidylcholine (Avanti) and monophosphoryl lipid A (Sigma-Aldrich). Female, 6-week-old MRL/lpr mice, maintained between 22–26 °C and 40%–60% humidity under a 12-h light cycle, were immunized three times at two-week intervals with 0.1 mg of the proteoliposome VMAT2 antigen. Single cells were harvested from mice spleens and were fused with NS-1 myeloma cells. To select antibodies that recognized the 3D structure of human VMAT2, we performed a multi-step screening method[40], using VMAT2, at each step, which included liposome-ELISA, denatured ELISA, and fluorescence size-exclusion chromatography. The collected clones were subsequently isolated by limiting dilution to establish monoclonal hybridoma cell lines. The resulting immunoglobulin-G (IgG0801) was purified with HiTrap Protein G HP (GE Healthcare) followed by proteolytic cleavage with papain (Nacalai Tesque). The Fab fragment (Fab0801) was then purified by size-exclusion chromatography (Superdex 200 10/300 GL, GE Healthcare) and affinity chromatography with a Protein A Sepharose 4 Fast-Flow column (GE Healthcare). The sequence of Fab0801 was

determined via standard 5′-RACE using total RNA isolated from hybridoma cells.

The sequence of the Fab0801 fragment light chain is provided below. CDR-L1, CDR-L2, and CDR-L3 are underlined.

DIVMTQSQKFMSTSVGDRVSITCKASQNVGTDVSWYQQKPGKSP KPLIYWASNRFTGVPDRFTGSRSGTDFTLTISNVQSEDLADYFCEQYSSY PLTFGAGTKLELKRADAAPTVSIFPPSSEQLTSGGASVVCFLNNFYPKDI NVKWKIDGSERQNGVLNSWTDQDSKDSTYSMSSTLTLTKDEYERHNSYT CEATHKTSTSPIVKSFNRNEC

The sequence of the Fab0801 fragment heavy chain is provided below. CDR-H1, CDR-H2, and CDR-H3 are underlined.

EVKLQESGAELVKPGASVKLSCKASGYTFTSYWIDWVKQRPGQGLE WIGNIYPGNSSTNYNEKFKNKATLTVDTSSSTAYMQLSSLTSDDSA VYYCAREDYYDGTYVYYAMDFWGQGTSVTVSSAKTTAP SVYPLAPVCGDTSGSSVTLGCLVKGYFPEPVTLTWNSGSLSSG VHTFPAVLQSDLYTLSSSVTVTSSTWPSQSITCNVAHPASSTKVDKK IEPRGPTIKPCPPCKCPAPNLLGGPSVFIFPPKIKDVLMISLSPIVTC VVVDVSEDDPDVQISWFVNNVEVHTAQTQTHREDYNSTLR VVSALPIQHQDWMSGKEFKC

### Cryo-EM grid preparation and data collection

The VMAT2-Fab0801 complex was prepared by mixing the purified VMAT2 and Fab0801 at a molar ratio 1:1.2 for 1 h on ice. The mixture was injected onto a Superdex 200 10/300 GL column (GE Healthcare), and the fractions containing the complex were concentrated to approximately 5 mg/ml for electron microscopy experiments. For the $VMAT2_{dop}$ and $VMAT2_{tet}$ samples, the complex was incubated with 2 mM dopamine or 0.1 mM tetrabenazine for 1 h on ice. The Quantifoil R1.2/1.3 holy carbon copper grid (Quantifoil) was glow-discharged at 7 Pa with 10 mA for 10 s using a JEC-3000FC sputter coater (JEOL) prior to use. A 3 µL aliquot was applied to the grid, blotted for 3.5 s with a blot force of 10 in 100% humidity at 8 °C, and plunged into liquid ethane using a Vitrobot Mark IV (Thermo Fisher Scientific). Cryo-EM data collection for screening sample quality and grid conditions was performed using a Glacios cryo-transmission electron microscope operated at 200 kV accelerating voltage with a Falcon4 camera (Thermo Fisher Scientific) at the Institute for Life and Medical Sciences, Kyoto University. After several screening sessions, data were collected using a Titan Krios (Thermo Fisher Scientific) at 300 kV accelerating voltage equipped with a direct K3 electron detector, Gatan BioQuantum energy filter (slit width of 20 eV) (Gatan), and Cs corrector (CEOS, GmbH), which were installed at the Institute for Protein Research, Osaka University. Data collection was carried out using SerialEM software[41] at a nominal magnification of ×81,000 (calibrated pixel size of 0.88 Å pixel⁻¹) with a total exposure time of 6.7 s (65 frames) with a defocus range of −0.8 to −1.8 µm. The detailed imaging conditions are described in Table 1.

### Cryo-EM data processing

All image processing was performed using Relion[42] and cryoSPARC[43]. The movie frames were aligned in 5 × 5 patches and dose-weighted in MotionCor2[44]. We manually inspected and curated the micrographs after contrast transfer function (CTF) estimation. The particles were selected by the Blob particle picker using a small image set, and two-dimensional (2D) classification, Ab initio reconstruction, heterogeneous refinement and nonuniform (NU) refinement were performed to yield templates for subsequent template picking. Template picking was used to repick the particles from all the micrographs. The picked particles were subjected to a 2D classification to discard particles in poorly defined classes. Multiple rounds of heterogeneous refinement were then performed against the ab initio models obtained from a subset of the micrographs in order to remove bad particles. The selected particles were extracted at full pixel size and subjected to Bayesian polishing[45] and NU refinement[46]. Resolutions were estimated using the "gold standard" criterion (FSC = 0.143). The local resolution

was calculated in cryoSPARC. Map sharpening was reevaluated with the Phenix autosharpen tool[47,48]. These maps were used for modeling. The processing strategy is described in Supplementary Fig. 3.

### Model building and refinement

The initial structural model of VMAT2 was generated by AlphaFold2[49] and our previous Fab fragment model (7DFP)[25] was used as the model for Fab0801. Model building was facilitated by using these initial structures. The VMAT2, Fab0801 models were manually built in Coot[50], followed by several rounds of real-space refinement using Phenix[51]. All molecular graphics were prepared using CueMol (http://www.cuemol.org) and UCSF ChimeraX[52]. The 3D reconstruction and model refinement statistics are summarized in Table 1.

### Transport assay

All mutant samples were prepared in the same manner as samples for cryo-EM. Transport assay was performed as described previously[53]. Aliquots of purified VMAT2 protein (20 µg) were mixed with asolectin liposomes (500 µg) and frozen in −80 °C for 10 min. The mixture was thawed quickly and diluted 60-fold with a reconstitution buffer containing 20 mM MES-KOH (pH 6.0), 5 mM magnesium acetate, and 0.1 M potassium chloride. Reconstituted proteoliposomes were obtained by centrifugation at 437,000 × g for 40 min at 4 °C with reconstitution buffer and then suspended in 0.2 mL of reconstitution buffer. A reaction mixture (130 µL) consisting of VMAT2 protein (0.4 µg) incorporated into proteoliposomes, 20 mM MES-KOH (pH 6.0) or 20 mM MOPS-KOH (pH 7.5), 0.1 M potassium chloride, 5 mM magnesium acetate, and 10 µM [³H]−5-hydroxytryptamine (0.5 MBq/µmol; PerkinElmer) or 10 µM [³H]-dopamine (0.5 MBq/µmol; PerkinElmer) was incubated for 1 min at 27 °C. The proteoliposomes were separated from the external medium using centrifuge columns containing Sephadex G-50 (fine) at 760 × g for 2 min at 4 °C to terminate the transport. Radioactivity in the eluate was measured with a liquid scintillation counter (PerkinElmer).

### Data analysis and statistics

All functional study data were analyzed using Prism (GraphPad) and are presented as mean ± standard error of the mean (SEM) of biologically independent experiments. Statistical analyses were performed using Prism (GraphPad) with one-way analysis of variance followed by Dunnett's test or two-tailed paired Student's $t$ test. Values with $P < 0.01$ are considered statistically significant.

### Reporting summary

Further information on research design is available in the Nature Portfolio Reporting Summary linked to this article.

## Data availability

The cryo-EM density maps and atomic coordinates have been deposited in the Electron Microscopy Data Bank (EMDB) and wwPDB under accession numbers EMD-37774 and 8WRD for the apo state of VMAT2-Fab0801 complex, EMD-37775 and 8WRE for the dopamine-bound VMAT2-Fab0801 complex, and EMD-37867 and 8WVG for the tetrabenazine-bound VMAT2-Fab0801 complex. Source data are provided with this paper.

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

## Acknowledgements

This work was supported by a Grant-in-Aid from the Japanese Ministry of Education, Culture, Sports, Science and Technology (19H00923 (S.I.) and 22KK0099, 23K06357 (D.I.)) and the Platform Project for Supporting Drug Discovery and Life Science Research (Basis for Supporting Innovative Drug Discovery and Life Science Research (BINDS)) from Japan Agency for Medical Research and Development (AMED) under the grant number JP21am0101079 and JP23ama121007 (S.I.), JP23ama121001 and 23809479 (D.I.). This study was also supported by the Takeda Science Foundation and the Mochida Memorial Foundation for Medical and Pharmaceutical Research (D.I.). This work was performed in part under the Collaborative Research Program as the Visiting Fellow of the Institute for Protein Research, Osaka University, VFCR–23-02 (S.I.) and in part using the cryo-electron microscope under the Collaborative Research Program of Institute for Protein Research, Osaka university, CEMCR-23-02 (H.A.). This work was also supported by the Cooperative Research Program (Joint Usage/Research Center Program) of the Institute for Life and Medical Sciences, Kyoto University.

## Author contributions

D.I. and S.I. designed the experiments. D.I. prepared the cryo-EM samples. D.I. and T.U. generated the antibody. D.I., M.J., J.K., Y. Sugita, T.N., T.K., and H.A. performed the cryo-EM analysis. D.I., T.U., and Y. Shiimura prepared the samples for the mutagenesis study. N.J. and T.M. performed the transport assay. D.I., M.J., and S.I. wrote the paper with assistance from all of the authors. D.I. and S.I. supervised the project. All authors have read and approved the final version of the manuscript.

## Competing interests

The authors declare no competing interests.
