## [Peer Review File · Nature Communications]

Neurotransmitter Recognition by Human Vesicular Monoamine Transporter 2REVIEWER COMMENTS

Reviewer #1 (Remarks to the Author):

The study by Im et al. describes cryoEM structures of VMAT2 in the apo state, in a substrate-bound state (dopamine) and an inhibitor-bound state (tetrabenazin). The structures are obtained using a construct with partial deletion of ECL1 and of the C-terminus, and with a bound Fab fragment derived from a monoclonal antibody targeting VMAT2. The solved structures reveal an architecture typical for the Major Facilitator Superfamily (MFS) transporter fold. The apo structure with dopamine bound adopts an outward-facing conformation while the tetrabenzin bound state adopts an occluded conformation. The structures allow the authors to describe in detail the binding sites of the two compounds. For dopamine, the binding site shows clear resemblance to the dopamine binding sites in the dopamine receptor and dopamine transporter. Tetrabenzine occupies the entire central cavity making interactions with residues both in the N- and C-terminal domains. In contrast, dopamine mainly binds residues in the C-terminal domain. Mutagenesis of key residues followed by uptake analysis on reconstituted transporter are used to substantiate the cryoEM findings. Finally, a possible transport mechanism is proposed based on predictions of pKa values for key Asp residues. Overall, this is a well-performed study that provides new insights into the structure and mechanism of VMAT2.

Specific points:

1. The novelty of the findings possibly could be better communicated. 2/3 of the abstract is background and the result is described in one not very informative sentence. Similarly, there is only “headline sentences” e.g in the last paragraph of the Introduction. Given that already one structure has been published of VMAT2, I think it would be worth emphasizing more precisely the significance of the present study.
2. I acknowledge that the authors mutate critical residues and analyze the mutants for uptake capacity upon reconstitution. However, a decrease in uptake does not necessarily reflect loss of transport capacity. It could as well reflect e.g. impaired folding leading to less functional protein. It is not clear how the authors correct for lower expression and/or possible structural instability of the mutants (which might lead to much less transporter in the liposomes compared to WT).
3. In line213, it is stated that Asp426 has a significantly higher pKa value compared to other acidic residues. When I look at Extended Data Figure 9, this is not so clear. I suggest modifying the description of the pKa predictions. Also, the reliability of the predictions must be detailly discussed given that they are so critical for the proposed mechanism.

Minor point: The irreversible VMAT2 inhibitor reserpine has been widely used to treat hypertension. I think it is worth mentioning that in the Introduction when describing VMAT2 as drug target.

Reviewer #2 (Remarks to the Author):

In this contribution by Im et al. three cryo-EM structures of human VMAT2 transporters in apo form, in complex with dopamine and in complex with tetrabenazine are reported. The structures are of a moderate resolution, but nevertheless of high quality and allow not only the analysis of general fold (which is unsurprisingly MFS) but also some analysis of the interactions with the substrate and the inhibitor in the binding site. Sadly the manuscript eventually goes into a complete speculation and in principle would require quite some additional data (functional, e.g. measurements of proton transport, and /or MD simulations) to prove the proposed transport mechanism and proton coupling, albeit I understand that it might be too troublesome to complete during the revision.

Some main points to be addressed for a further consideration of this manuscript for a publication:

- in all transport assays, there is no control shown. Please include empty liposomes as a control;
- in the complex with the dopamine, Phe334 is not properly modelled into the density, perhaps affecting the structural figures and calculations of tunnels;
- in the complex with the inhibitor, there is a spherical patch of density between the inhibitor and Lys138 which is either water or an ion, so it should be taken into consideration as well;
- the comparison with the published Human VMAT2 in complex with serotonin should be more pronounced and the SI figure 8 should be moved to the main text;
- add more information about ECL1 loop which was excluded from the construct, despite it seems not very important for transport, it might be important for regulation;

minor issues:

- line 84, 'result' should be 'results'
- line 171 this is not an electron density
- line 187 what is meant by 'meaningful interactions'?
- line 205 if it is indeed Asn146 mutated then it should be N146Q and not D146Q
- fig4, the mutation should be Q142N and not N142N
- line 260, replace 'illuminated' with 'highlighted'
- line 297, replace 'reported in our previous paper' to 'reported previously'

-line 319, space is missing in 150mM NaCl

-line 386 'revaluted'? Do you mean re-evaluated?

For figures with the electrostatic potential calculations, please indicate the values +/- X kT/e-

-fig4 panel c caption - do you mean orange sticks?

-fig3 panel f and fig 2 panel e, indicate the distances

-fig 2 panel d and fig 3 panel d, please avoid the combination of red and green

Reviewer #3 (Remarks to the Author):

Dear Im, Jormakka, and Juge et al,

I enjoyed reading and reviewing your manuscript for publication in Nature Communications. Despite the first structures of human VMAT2 very recently published in Nature in November of 2023 by Pidathala et al, your exceptional work of resolving the structure of this protein in the outward facing apo and dopamine-bound, as well as occluded tetrabenazine conformation clearly enhances the knowledge on this transporter. Extensive mutagenesis and transport assays greatly complement your study and together provide a model for how this transporter works.

I do have a few concerns and recommend revisions and will need clarifying information to make a recommendation for publication:

My concerns:

a) You mention that the Fab fragment does not notably affect the transport assay but from the methods it was not clear to me how much Fab was added and what your estimate is on how many transporters are actually bound by the Fab. Based on the single-particle processing it looks like not all transporters have a Fab bound since only about 1.47 million of 23.4 million apo-particles look like they have a clear Fab density, which is only about 6%. Could it be that the activity you measure came from transporters without Fabs bound? Including a SDS-gel of your SEC fractions you pooled for cryo or potentially also the reconstitution for the transport assay would help. Otherwise, I suggest running gels of your proteoliposomes and analyzing the transporter to Fab ratio. Native gels can also help to indicate how much free transporter vs. Fab bound transporters you have in your sample. Was the Fab added after reconstitution or was the Fab-bound sample used for reconstitution. Please clarify.

b) I am a structural biology expert and not a functional experts. Reading the methods on the transport assay and then looking at all the figures does not make me feel confident that I could repeat this experiment. In the methods text in line 408 it says "At the indicated time points,...": I

have not seen any indicated time points. Are there figures missing? Also how is driving force created and what is +/- driving force?

c) You mention the recently published human VMAT2-5HT structure but not the others: Pidathala et al also solved the structure of VMAT2 in complex with tetrabenazine. Please compare experimental setup and results. Even if the results are very similar, it is good to mention that two independent groups came to the same structure.

d) Pidathala et al also resolved VMAT2 in complex with reserpine. Please mention and compare.

e) Towards the end of the introduction it is mentioned that homology models and mutagenesis experiments have proposed the location of the substrate binding site and transport mechanism. Please discuss these and mention what was predicted correctly and what is now more clear after solving the structures.

f) You mention salt bridges throughout the manuscript but to my knowledge salt bridges are formed between residues of opposite charges like Glu-Arg, not Asp-Asn. Rename to polar or electrostatic interactions?

g) deltaC removing 34 residues is not really indicated in SI figure 1. Maybe make that more clear in the figure or the legend.

h) Please clarify why a His-tag purification was done with a FLAG tagged protein. At some point a his-tagged GFP was mentioned but that is not clear from S2a.

i) The apo and Dop structures were resolved in DDM/CHS, the tet structure in LMNG. Why the change, where is the SEC profile in LMNG and do you think the conformation is influenced by the type of detergent?

j) Please add SDS-gels and if you have Native-gels of your purifications, gelfiltration fractions, some of which you pooled for cryo-EM, possibly transport assays.

k) I did not see pdb and map files to review but the validation reports look good to me besides one thing I noticed; I would check the ligand XEQ which got a high Z score. Please address. It is nice to see that the Fab was modeled as well, not just the transporter. Adding the Fab-transporter interaction interface to figure S5 or elsewhere would be of interest.

l) Please add the dose rate to table 1 or the methods.

m) Please add extraction box sizes in methods and workflow figures or legends.

n) From figure S3 it is not clear to me which steps were performed in RELION and which in cryoSPARC. Please clarify.

o) Please add a scale bar to all micrograph images.

Minor points:

1. I noticed you did not perform Bayesian polishing with the “tet” sample. Why not?
2. In many figures dashed lines between residues are indicated without distances. Please add the distances in the figure or legend.
3. Line 179: check wording. Maybe “... with residues on both of the N-terminal and ...”?
4. Please check figure 4d: N142N? Do you mean Q142N?
5. Line 295: reference to S1, did you mean S2a?
6. Line 347-349: gel filtration followed by affinity or the other way around?
7. Line 360: no space between number 8 and degree sign.
8. Fix Fig 4 figure legend.
9. Line 614: remove “of” after wild-type?
10. SI4 figure legend, please fix. I guess c and d in line 625 are meant to be e and f.
11. Line 631: indicate instead of indicated.
12. SI references for all the pDBs mentioned would be nice.

Reviewer #4 (Remarks to the Author):

The work presented by Im and colleagues offers structural insight into the human VMAT2 protein, a member of the SLC18 family. This secondary active transporter mediates the uptake of cationic monoamine neurotransmitters into synaptic vesicles in exchange for protons. As the authors pointed out, VMAT2 is a pharmaceutical target for several neurodegenerative and psychiatric diseases, such as Huntington's disease. This highlights the significance of the protein and, consequently, the research.

The authors present three high-resolution cryo-EM structures of detergent-solubilized VMAT2 in complex with a Fab fragment in the apo, substrate-bound, and inhibitor-bound states. The apo and dopamine-bound structures reveal the outward-open conformation of the transporter. The structural data are supported by functional analysis, mutagenesis experiments, and the prediction of pKa values for essential residues. The data have been used to propose a VMAT2-mediated transport mechanism. This is an elegant manuscript, featuring well-written text, clear and informative figures, and a comprehensive description of the methods employed for the research. Notably, the cryo-EM maps are well-resolved, and the atomic models of the structures are of high quality (with a single concern listed below).

As of today, this work provides novel insights into our understanding of the VMAT2 protein. However, there are several concerns that should be addressed, with the majority of them related to the proposed transport mechanism. In addition, a very recent article by Pidathala et al., Nature, 2023, raises additional considerations. While the authors acknowledged this publication in [Ref. 27], the results presented in [Ref. 27] shed new light on the story presented here by Im et al. Therefore, in my opinion, a single paragraph and one extended figure (lines 154-165, Extended Data Fig. 8) in which both sets of results are compared and discussed are not sufficient.

Major comments:

1. My main concern is that the authors captured the substrate-bound state in the outward-open conformation, while [Ref. 27] reported the substrate-bound state in the inward-open conformation. This is surprising and, at the same time, exciting, but it may have implications for the proposed transport mechanism and should therefore be elaborated further. Is it possible that the VMAT2 sample was prepared differently in a way that affected the favored conformation, for example, in a detergent micelle versus in a lipid-filled nanodisc?

2. The authors report the tetrabenazine-bound structure of VMAT2. In this inhibitor-bound state, the transporter adopts a distinct, occluded conformation. The authors also claim (line 276): “However, the occluded structure indicates that there are significant structural changes occurring around the substrate binding site during the transition to inward-facing conformation”. At the same time, the occluded state is not included in the proposed transport mechanism (Figure 5). This is contradictive. My question is whether the occluded state is part of the transport cycle or if it is “outside” the cycle?

3. In Figure 5, panels i)-iv) are well-supported by the data presented in the manuscript. However, the remaining panels are speculative and could benefit from further support from the data published in [Ref.27]. The authors claim that the transition from the inward- to the outward-open conformation “could be catalyzed through an induced fit by the substrate, facilitated by the deprotonated Asp399”. This implies that state viii) is energetically unfavored, yet it has been recently captured [Ref.27]. Additionally, instead of assuming that in the inward-open state “Asp399 and Asp426 [...] have more normal pKa values (i.e. pKa ~4)” (panels vi)-vii)), the authors could perform a similar pKa prediction using the structure in the inward-open conformation.

4. The proton coupling analysis is an interesting part of the manuscript that rationalizes the counter-transport of proton and monoamine in a 2:1 stoichiometry. The interaction networks of

Gln142-Asp426-Asn146 and Asp399-Asn305 can also be analyzed in the inward-open conformation of VMAT2 to further support, or even improve, the proposed transport mechanism.

5. All three atomic models were built very carefully and show good validation statistics. However, please correct the F334 residue in the dopamine-bound model, as the side chain is positioned outside obvious density. Given its proximity to the substrate molecule, this may affect the characterization of the substrate binding pocket.

Minor comments:

6. I would suggest omitting the statement from the abstract: “because the structure of VMAT2 is still unknown”.

7. The statistical analysis of the functional data is missing, which is particularly crucial for results such as the D399N mutant, where the authors claim to have observed “reduced transport activity”.

8. Extended Data Figure 8 - please specify which parts of the models were superimposed. Given that these two structures present two distinct conformations, please consider aligning them precisely in register to the region of the binding pocket.

9. Please provide the sequence of the generated Fab or the CDRs of the Fab in the text.

10. The angular distribution of particles used in the final reconstructions is missing.

11. L. 408 “At the indicated time points” – please specify the time points for each experiment.

12. L. 243 – “released” instead of “transported” could be less confusing.

13. Figure 4d – “Q142N” instead of “N142N”.

14. Extended Data Figure 9 – “residue” instead of “resudue”.

REVIEWER COMMENTS

Below we provide our replies to the reviewers' comments in blue, following the reviewers' remarks in black.

Reviewer #1 (Remarks to the Author):

The study by Im et al. describes cryoEM structures of VMAT2 in the apo state, in a substrate-bound state (dopamine) and an inhibitor-bound state (tetrabenazin). The structures are obtained using a construct with partial deletion of ECL1 and of the C-terminus, and with a bound Fab fragment derived from a monoclonal antibody targeting VMAT2. The solved structures reveal an architecture typical for the Major Facilitator Superfamily (MFS) transporter fold. The apo structure with dopamine bound adopts an outward-facing conformation while the tetrabenzin bound state adopts an occluded conformation. The structures allow the authors to describe in detail the binding sites of the two compounds. For dopamine, the binding site shows clear resemblance to the dopamine binding sites in the dopamine receptor and dopamine transporter. Tetrabenzine occupies the entire central cavity making interactions with residues both in the N- and C-terminal domains. In contrast, dopamine mainly binds residues in the C-terminal domain. Mutagenesis of key residues followed by uptake analysis on reconstituted transporter are used to substantiate the cryoEM findings. Finally, a possible transport mechanism is proposed based on predictions of pKa values for key Asp residues. Overall, this is a well-performed study that provides new insights into the structure and mechanism of VMAT2.

Thank you for your interest in reading and reviewing our paper. We will respond to the specified individuals regarding the items you highlighted.

Specific points:

1. The novelty of the findings possibly could be better communicated. 2/3 of the abstract is background and the result is described in one not very informative sentence. Similarly, there is only “headline sentences” e.g in the last paragraph of the Introduction. Given that already one

structure has been published of VMAT2, I think it would be worth emphasizing more precisely the significance of the present study.

Thank you for your suggestion. In response, We have revised the abstract and introduction to highlight the novelty and significance of this study (Lines 42-44, 77-79, 83-86).

2. I acknowledge that the authors mutate critical residues and analyze the mutants for uptake capacity upon reconstitution. However, a decrease in uptake does not necessarily reflect loss of transport capacity. It could as well reflect e.g. impaired folding leading to less functional protein. It is not clear how the authors correct for lower expression and/or possible structural instability of the mutants (which might lead to much less transporter in the liposomes compared to WT).

Thank you for your valuable comments. The mutant proteins used in this study were recombinantly expressed and purified. The stability and yield of the purified proteins were assessed by measuring protein concentration (nano-drop) and SDS-PAGE prior to liposome reconstitution. Although the expression levels of each mutant protein were not corrected for in the purification, we obtained approximately 60 µg to 300 µg per 1 L of cell culture for all mutant proteins. Purified protein samples were concentrated to similar levels before proteoliposome preparation, as such we believe variations in expression levels were minimised in the proteoliposome level. The results of SDS-PAGE after purification for all samples have been added in Fig. S2d.

3. In line213, it is stated that Asp426 has a significantly higher pKa value compared to other acidic residues. When I look at Extended Data Figure 9, this is not so clear. I suggest modifying the description of the pKa predictions. Also, the reliability of the predictions must be detailly discussed given that they are so critical for the proposed mechanism.

Thank you for pointing this out. To clarify this, we have adjusted these expressions in the manuscript (Lines 223-225). Additionally, we have included a pKa prediction for serotonin-bound VMAT2 (Pidathala et al., PDB 8T6B) of the inward-facing structure in Fig. S9. The predicted pKa value is smaller than that of Asp426 in the outward-facing structure, supporting the proposed

transport mechanism of VMAT2 in this paper. A note regarding this has been added to the manuscript (Lines 233-238).

Minor point: The irreversible VMAT2 inhibitor reserpine has been widely used to treat hypertension. I think it is worth mentioning that in the Introduction when describing VMAT2 as drug target.

Thank you for your suggestion. We have edited the introduction by noting that reserpine is widely utilized for hypertension treatment, targeting VMAT2. Proper citations have also been provided (Lines 74-75).

Reviewer #2 (Remarks to the Author):

In this contribution by Im et al. three cryo-EM structures of human VMAT2 transporters in apo form, in complex with dopamine and in complex with tetrabenazine are reported. The structures of a moderate resolution, but nevertheless of high quality and allow not only the analysis of general fold (which is unsurprisingly MFS) but also some analysis of the interactions with the substrate and the inhibitor in the binding site. Sadly the manuscript eventually goes into a complete speculation and in principle would require quite some additional data (functional, e.g. measurements of proton transport, and /or MD simulations) to prove the proposed transport mechanism and proton coupling, albeit I understand that it might be too troublesome to complete during the revision.

Thank you for reviewing our paper. Your constructive suggestions are greatly appreciated for enhancing its quality. We will provide responses to the relevant individuals regarding your points.

Some main points to be addressed for a further consideration of this manuscript for a publication:

1. In all transport assays, there is no control shown. Please include empty liposomes as a control;

Thank you for bringing this matter to our attention. As correctly noted, the control was absent in this activity measurement. We apologize for the omission. We have rectified this by reintroducing the transport activity of 5-HT and dopamine in the liposome back as a control (Figs. S6a, b).

2. In the complex with the dopamine, Phe334 is not properly modelled into the density, perhaps affecting the structural figures and calculations of tunnels;

We appreciate your observation. Following your feedback, we conducted a detailed examination of the model and map of Phe334 of VMAT2_{dop}. Indeed, as you pointed out, the side chain of Phe334 was found to be outside the map. Consequently, we made necessary modifications to the structure and carried out refinements accordingly. The corresponding figure has been replaced with the updated model.

3. In the complex with the inhibitor, there is a spherical patch of density between the inhibitor and Lys138 which is either water or an ion, so it should be taken into consideration as well;

We appreciate your feedback. Following your observation, we detected a water-like density between the inhibitor and Lys138. Subsequently, we conducted a refinement with a water molecule and updated the model accordingly. However, it's pertinent to mention that this water molecule does not participate in hydrogen bonding with the inhibitor and does not appear to significantly influence ligand binding. Therefore, we have opted not to depict it in the figures.

4. The comparison with the published Human VMAT2 in complex with serotonin should be more pronounced and the SI figure 8 should be moved to the main text;

Thank you for your valuable feedback. In accordance with your suggestion, we have conducted a more thorough comparison with the serotonin-bound VMAT2 (PDB 8T6B) (Figs. S8, S10). Moreover, conducting pKa prediction of this structure has provided additional details in regards to the proton coupling and conformational change mechanism posited in this paper (Fig. S9). These changes are detailed in Lines 233-238.

5. Add more information about ECL1 loop which was excluded from the construct, despite it seems not very important for transport, it might be important for regulation;

We appreciate your observation. As correctly pointed out, the ECL1 loop's function is not crucial for transport but could be significant in regulating transporter function. A note has been added to the manuscript to address this (Lines 129-130).

Minor issues:

1. Line 84, 'result' should be 'results'

Acknowledging your suggestion, we have rectified the pertinent section of the text (Line 91).

2. Line 171 this is not an electron density

Thank you for your feedback. We have addressed the incorrect word usage by replacing it with the appropriate term. (Line 179)

3. Line 187 what is meant by 'meaningful interactions'?

Regrettably, our previous phrase was ambiguous. We intended to denote an interaction between the dimethoxy group and the surrounding residues. To eliminate ambiguity, we have substituted the text with more appropriate wording (Lines 195-196).

4. Line 205 if it is indeed Asn146 mutated then it should N146Q and not D146Q

We appreciate your feedback. It was noted that the variant should be designated as N146Q, not D146Q as previously indicated. This has been updated accordingly (Line 216).

5. Fig4, the mutation should be Q142N and not N142N

We regret any confusion caused. The appropriate mutant designation is Q142N, not N142N. This has been addressed and amended in Fig. 4.

6. Line 260, replace 'illuminated' with 'highlighted'

We appreciate your clarification. The identified section has been modified to 'highlighted' as specified (Line 283).

7. Line 297, replace 'reported in our previous paper' to 'reported previously'

We appreciate your suggestion. The indicated section has been amended to 'reported previously' for clarity (Line 311).

8. Line 319, space is missing in 150mM NaCl

Appreciate the catch. An extra space has been inserted as indicated (Line 333).

9. Line 386 'revaluted'? Do you mean re-evaluated?

The pertinent section has been adjusted to 'reevaluated' as suggested (Line 414).

10. For figures with the electrostatic potential calculations, please indicate the values +/- X kT/e-

Your suggestion has been duly noted. The electrostatic potential figure has been updated to incorporate the value of $\pm X$ kT/e- (Figs 1e, f, 2a, b, 3a, b).

11. Fig4 panel c caption - do you mean orange sticks?

Confirmed, they are. The mention of color was incorrect. Orange has been appended as necessary (Line 640).

12. Fig3 panel f and fig 2 panel e, indicate the distances

We acknowledge your recommendation. The distances between atoms have been measured and showcased in the pertinent figures (Figs. 2e, 3f).

13. Fig 2 panel d and fig 3 panel d, please avoid the combination of red and green

We appreciate your comment and are cognizant of the importance of considering individuals with color vision deficiencies. However, the extensive array of colors used in the panels poses challenges in entirely avoiding green/red combinations. Nevertheless, for the final figure version, we will explore options to mitigate this issue.

Reviewer #3 (Remarks to the Author):

Dear Im, Jormakka, and Juge et al,

I enjoyed reading and reviewing your manuscript for publication in Nature Communications. Despite the first structures of human VMAT2 very recently published in Nature in November of 2023 by Pidathala et al, your exceptional work of resolving the structure of this protein in the outward facing apo and dopamine-bound, as well as occluded tetrabenazine conformation clearly enhances the knowledge on this transporter. Extensive mutagenesis and transport assays greatly complement your study and together provide a model for how this transporter works. I do have a few concerns and recommend revisions and will need clarifying information to make a recommendation for publication:

We extend our gratitude for reviewing our paper. We will provide responses to your questions individually.

My concerns:

a) You mention that the Fab fragment does not notably affect the transport assay but from the methods it was not clear to me how much Fab was added and what your estimate is on how many transporters are actually bound by the Fab. Based on the single-particle processing it looks like not

all transporters have a Fab bound since only about 1.47 million of 23.4 million apo-particles look like they have a clear Fab density, which is only about 6%. Could it be that the activity you measure came from transporters without Fabs bound? Including a SDS-gel of your SEC fractions you pooled for cryo or potentially also the reconstitution for the transport assay would help. Otherwise, I suggest running gels of your proteoliposomes and analyzing the transporter to Fab ratio. Native gels can also help to indicate how much free transporter vs. Fab bound transporters you have in your sample. Was the Fab added after reconstitution or was the Fab-bound sample used for reconstitution. Please clarify.

We appreciate your feedback. Initially, in the methodology section, we detailed the binding ratio of VMAT2 to Fab (Lines 381-382). Subsequently, this same sample was utilized for the reconstitution of the proteoliposome. We also consider that not all Fab fragments are bound to VMAT2, as you highlighted. Nonetheless, as evidenced in Extended Fig. 2c, the peak position with/without Fab exhibits significant variation depending on the SEC pool, and the binding of Fab is convincingly demonstrated by each SDS-PAGE Gel. Accordingly, we will include the SDS-PAGE gel picture in Fig. S2d.

b) I am a structural biology expert and not a functional experts. Reading the methods on the transport assay and then looking at all the figures does not make me feel confident that I could repeat this experiment. In the methods text in line 408 it says “At the indicated time points,...”: I have not seen any indicated time points. Are there figures missing? Also how is driving force created and what is +/- driving force?

Thank you for your comment. The transport assay was performed at a time point of 1 min, and we have changed the contents of Method - Transport assay (Line 436). Also, we have added the details of driving force in figure legend of Fig 1c (Lines 614-615).

c) You mention the recently published human VMAT2-5HT structure but not the others: Pidathala et al also solved the structure of VMAT2 in complex with tetrabenazine. Please compare experimental setup and results. Even if the results are very similar, it is good to mention that two independent groups came to the same structure.

We appreciate your suggestion. It is noteworthy that the same ligand-bound structure was tested using independent methods, yielding similar results, as you mentioned. We have compared the tetrabenazine-bound structure of Pidathala et al. (Ref. 17) and integrated the details into the manuscript and figures (Lines 197-200, Supplementary Figs. 8c, d).

d) Pidathala et al also resolved VMAT2 in complex with reserpine. Please mention and compare.

Thank you for your observation. As you mentioned, the Pidathala et al. determined the structure of reserpine-bound VMAT2. Shortly thereafter, the same inhibitor-bound structure was published by the Wu et al. (Ref. 18). In contrast to the Pidathala et al. structure, the Wu et al. structure does not contain the Y418S mutation, thus allowing for a more straightforward discussion of the structural changes in the transporter. Accordingly, we have cited this structure for further comparison and discussion (Figs. S8e, f, S10, Lines 271-279, 292-299).

e) Towards the end of the introduction it is mentioned that homology models and mutagenesis experiments have proposed the location of the substrate binding site and transport mechanism. Please discuss these and mention what was predicted correctly and what is now more clear after solving the structures.

Thank you for your comment. I apologize for the lack of explanation. The relevant details have already been mentioned in the result section (Lines 114-117, 218-220). Yaffe et al. (Ref. 14, 15) inferred the location of inhibitor binding and clusters important for conformational changes in rat VMAT2 through a combination of homology modeling and mutant experiments. Our cryo-EM structure confirmed the presence of intracellular gate and hydrogen bond clusters, indicating that their predictions were accurate. The structural information in our paper, along with the associated protonation predictions, could suggest an additional stepwise transport mechanism for VMAT2, complementing the claims of Yaffe et al.

f) You mention salt bridges throughout the manuscript but to my knowledge salt bridges are

formed between residues of opposite charges like Glu-Arg, not Asp-Asn. Rename to polar or electrostatic interactions?

Thank you for pointing this out. As you mentioned, salt bridges form between residues with opposite charges, such as Glu and Arg. We have corrected the descriptions of these interactions (Lines 256, 257, 266).

g) deltaC removing 34 residues is not really indicated in SI figure 1. Maybe make that more clear in the figure or the legend.

Thank you for pointing this out. The details of the deltaC location are provided in Fig S2a, and We have indicated in the figure legend that the secondary structure in Fig. S1 represents VMAT2 for cryo-EM with deltaC (Lines 654-655).

h) Please clarify why a His-tag purification was done with a FLAG tagged protein. At some point a his-tagged GFP was mentioned but that is not clear from S2a.

Thank you for your comment. The N-terminus of the VMAT2 construct used in this study contains a FLAG tag, which was added to confirm the cellular expression level. We used the His tag on the C-terminal side during purification. This information was omitted in Fig. S2a, so we have corrected it. The C-terminal GFP was utilized during high-throughput screening of the constructs and to confirm purity at each purification step, as described in Methods - Protein expression and purification. GFP was removed by TEV protease treatment prior to SEC and was not included in the final product.

i) The apo and Dop structures were resolved in DDM/CHS, the tet structure in LMNG. Why the change, where is the SEC profile in LMNG and do you think the conformation is influenced by the type of detergent?

Thank you for pointing this out. There is essentially no conformational change among different detergent types. However, we did observe a shift in elution position depending on the detergent size. We have included the SEC profile in Fig. S2c.

j) Please add SDS-gels and if you have Native-gels of your purifications, gelfiltration fractions, some of which you pooled for cryo-EM, possibly transport assays.

Thank you for your suggestion. We have added SDS-PAGE gels of the samples for cryo-EM and the transport assay in Fig. S2d.

k) I did not see pdb and map files to review but the validation reports look good to me besides one thing I noticed; I would check the ligand XEQ which got a high Z score. Please address. It is nice to see that the Fab was modeled as well, not just the transporter. Adding the Fab-transporter interaction interface to figure S5 or elsewhere would be of interest.

Thank you for your comment. We utilized a racemic mixture of tetrabenazine in the experiment, and despite efforts to refine it using a specific chiral dictionary file, we were unable to resolve it. Given that the ligand model did not significantly deviate from the map, we propose to adopt the current model as is. The interaction interface of the Fab and transporter has been added to Fig. S5c, d.

l) Please add the dose rate to table 1 or the methods.

Thank you for your suggestion. The dose rate has been included in Table 1 as requested.

m) Please add extraction box sizes in methods and workflow figures or legends.

Thank you for your suggestion. The extraction box size has been added to Fig. S3 as requested.

n) From figure S3 it is not clear to me which steps were performed in RELION and which in cryoSPARC. Please clarify.

Thank you for your comment. Fig. S3 has been revised to incorporate the software actually used in each step as requested.

o) Please add a scale bar to all micrograph images.

Thank you for your feedback. We have added a scale bar to all micrograph images (Fig. S3).

Minor points:

1. I noticed you did not perform Bayesian polishing with the “tet” sample. Why not?

Thank you for your comment. Although there is no specific reason, the tetrabenazine map was clearly identified in map without Bayesian polishing, leading us to select this map as the final one.

2. In many figures dashed lines between residues are indicated without distances. Please add the distances in the figure or legend.

Thank you for your suggestion. We have added the distances in the required figures.

3. Line 179: check wording. Maybe “... with residues on both of the N-terminal and ...”?

Thank you for your comment. The wording in the relevant section has been revised (Line 187).

4. Please check figure 4d: N142N? Do you mean Q142N?

Q142N is correct, as you indicated. The relevant manuscript section has been revised accordingly (Fig. 4d).

5. Line 295: reference to S1, did you mean S2a?

Thank you for pointing this out. My apologies, Fig. S2a is correct. The necessary changes have been made (Line 309).

6. Line 347-349: gelfiltration followed by affinity or the other way around?

Thank you for your comments. The method described in the original manuscript is valid. Following papain treatment during antibody purification, the Fab and Fc regions are separated by SEC. The remaining Fc is then removed using a Protein A column to isolate the final Fab region.

7. Line 360: no space between number 8 and degree sign.

Thank you for your suggestion. The space between the number and the degree sign has been removed (Line 388).

8. Fix Fig 4 figure legend.

Thanks for pointing this out, we have changed the Fig. 4 legend for clarity (Lines 639-640).

9. Line 614: remove "of" after wild-type?

Thank you for your suggestion. I have removed the "of" in the relevant section (Line 662).

10. SI4 figure legend, please fix. I guess c and d in line 625 are meant to be e and f.

Thank you for pointing this out. There was an indication error. The legends in Fig. S4 have been corrected to ensure they align accurately with the figure (Line 677).

11. Line 631: indicate instead of indicated.

We couldn't find such content in this manuscript. I meticulously examined all occurrences of 'indicate' in the legend of Fig. S5 and confirmed their accuracy.

12. SI references for all the pDBs mentioned would be nice.

Thank you for your suggestion. I have added references to all PDB's mentioned in the manuscript.

Reviewer #4 (Remarks to the Author):

The work presented by Im and colleagues offers structural insight into the human VMAT2 protein, a member of the SLC18 family. This secondary active transporter mediates the uptake of cationic monoamine neurotransmitters into synaptic vesicles in exchange for protons. As the authors pointed out, VMAT2 is a pharmaceutical target for several neurodegenerative and psychiatric diseases, such as Huntington's disease. This highlights the significance of the protein and, consequently, the research.

The authors present three high-resolution cryo-EM structures of detergent-solubilized VMAT2 in complex with a Fab fragment in the apo, substrate-bound, and inhibitor-bound states. The apo and dopamine-bound structures reveal the outward-open conformation of the transporter. The structural data are supported by functional analysis, mutagenesis experiments, and the prediction of pKa values for essential residues. The data have been used to propose a VMAT2-mediated transport mechanism. This is an elegant manuscript, featuring well-written text, clear and informative figures, and a comprehensive description of the methods employed for the research. Notably, the cryo-EM maps are well-resolved, and the atomic models of the structures are of high quality (with a single concern listed below).

As of today, this work provides novel insights into our understanding of the VMAT2 protein. However, there are several concerns that should be addressed, with the majority of them related to the proposed transport mechanism. In addition, a very recent article by Pidathala et al., Nature, 2023, raises additional considerations. While the authors acknowledged this publication in [Ref. 27], the results presented in [Ref. 27] shed new light on the story presented here by Im et al. Therefore, in my opinion, a single paragraph and one extended figure (lines 154-165, Extended

Data Fig. 8) in which both sets of results are compared and discussed are not sufficient.

→ Thank you very much for reviewing our paper. We will respond to each of your questions below.

Major comments:

1. My main concern is that the authors captured the substrate-bound state in the outward-open conformation, while [Ref. 27] reported the substrate-bound state in the inward-open conformation. This is surprising and, at the same time, exciting, but it may have implications for the proposed transport mechanism and should therefore be elaborated further. Is it possible that the VMAT2 sample was prepared differently in a way that affected the favored conformation, for example, in a detergent micelle versus in a lipid-filled nanodisc?

Thank you for your comment. As you mentioned, Pidathula et al. (Ref. 17) presented the structure of the inward-facing conformation of the setoroin-bound VMAT2. While it is possible that the conformation of VMAT2 could change depending on the experimental conditions, we believe in this case the introduction of a point mutation (Y418S) was critical for obtaining this structure. The Y418S mutant appears to disrupt the formation of the intracellular gate, thus “locking” the protein in an inward-facing conformation. Indeed, this provides further anecdotal support for our proposed mechanism. Specifically, we proposed in our model that the long-range conformational changes following the protonation of D426 involve the disruption of the R217-D411-Y418 triad, leading to outward-to-inward conformational change. Conversely, the inability for this triad to re-form likely “locks” the protein in the inward facing conformation. To clarify and elaborate on this, we have amended the following section (Lines 271-279) and also included new figures (Fig. S10).

2. The authors report the tetrabenazine-bound structure of VMAT2. In this inhibitor-bound state, the transporter adopts a distinct, occluded conformation. The authors also claim (line 276): “However, the occluded structure indicates that there are significant structural changes occurring around the substrate binding site during the transition to inward-facing conformation”. At the same time, the occluded state is not included in the proposed transport mechanism (Figure 5). This is

contradictive. My question is whether the occluded state is part of the transport cycle or if it is “outside” the cycle?

Thank you for your comment. We have included the occluded states in the Fig. 5 to clarify the transport mechanism schematic. While transporters undergo energetically significant yet transient occluded states, we cannot be certain that the detailed observations made in the tetrabenazine-bound structure reflect those occurring for the physiological substrate, particularly around the substrate site. However, it is plausible that the global structural changes represent one of the occluded states in the transport cycle. Accordingly, we have referenced this in the schematic of the proposed mechanism.

3. In Figure 5, panels i)-iv) are well-supported by the data presented in the manuscript. However, the remaining panels are speculative and could benefit from further support from the data published in [Ref.27]. The authors claim that the transition from the inward- to the outward-open conformation “could be catalyzed through an induced fit by the substrate, facilitated by the deprotonated Asp399”. This implies that state viii) is energetically unfavored, yet it has been recently captured [Ref.27]. Additionally, instead of assuming that in the inward-open state “Asp399 and Asp426 [...] have more normal pKa values (i.e. pKa ~4)” (panels vi)-vii)), the authors could perform a similar pKa prediction using the structure in the inward-open conformation.

Thank you for your comment. This is related to what we answered in question 1. As previously noted, Pidathula et al. presented a structure of a serotonin-bound inward-facing conformation of VMAT2, achieved through a point mutation (Y418S) that “locked” the protein in this conformation.

To expand on this, an interesting observation is that the reserpine bound inward-facing conformation from both Pidathala et al (PDB 8T6B) and Wu et al. (PDB 8JTC) are virtually identical (RMSD 0.45 Å over 335 Cα atoms), while comparing these structures to the serotonin-bound structure (PDB 8T6A) illustrates small but significant differences. In particular, the intracellular halves of TMs 5, 8, and 10 have shifted towards the centre of the protein (Fig. S10d).

As such, the serotonin-bound structure appears more “closed” than the reserpine structures, which we speculate is a step towards a closure of the intracellular gate and formation of an occluded state. This may be facilitated by the mentioned deprotonation or substrate induced fit (or combination of the two). However, due to the Y418S mutation, the intracellular gate is unable to fully close.

We have furthermore, as suggested by the reviewer, included included a pKa prediction for serotonin-bound inward-facing VMAT2 (Pidathala et al., PDB 8T6B) in Fig. S9. The predicted pKa value for Asp426 is lower than that of the outward-facing structure, supporting the proposed transport mechanism of VMAT2 in this paper (in particular when considering the local pH). A note regarding this has been added to the manuscript (Lines 233-238).

4. The proton coupling analysis is an interesting part of the manuscript that rationalizes the counter-transport of proton and monoamine in a 2:1 stoichiometry. The interaction networks of Gln142-Asp426-Asn146 and Asp399-Asn305 can also be analyzed in the inward-open conformation of VMAT2 to further support, or even improve, the proposed transport mechanism.

We have, as suggested by the reviewer, included the inward-facing conformation in our analysis of the transport mechanism, as noted above. Of note is that the structures are likely in their deprotonated states, given the pH of the buffers used in the protein preparations (pH 7.5-8.0).

5. All three atomic models were built very carefully and show good validation statistics. However, please correct the F334 residue in the dopamine-bound model, as the side chain is positioned outside obvious density. Given its proximity to the substrate molecule, this may affect the characterization of the substrate binding pocket.

Thank you for your suggestion. The side chain model of F334 in VMAT2_{dop} has been corrected, and the updated structure has been re-registered in the Protein Data Bank (PDB).

Minor comments:

6. I would suggest omitting the statement from the abstract: “because the structure of VMAT2 is still unknown”.

Thank you for pointing this out. The relevant section has been removed from the abstract.

7. The statistical analysis of the functional data is missing, which is particularly crucial for results such as the D399N mutant, where the authors claim to have observed “reduced transport activity”.

Thank you for your suggestion. New significance tests have been conducted for all transport activity experiments presented in this paper, and the figures have been updated accordingly to illustrate the results (Figs. 1c, 2f, 4d, S6). And we have also updated the method section (Lines 441-455).

8. Extended Data Figure 8 - please specify which parts of the models were superimposed. Given that these two structures present two distinct conformations, please consider aligning them precisely in register to the region of the binding pocket.

Thank you for your comment. Fig. S8b will now depict an enlargement of the area highlighted in Fig. S8a. Additionally, we have substituted it with a revised figure superimposed on the ligand binding site to accommodate conformational changes.

9. Please provide the sequence of the generated Fab or the CDRs of the Fab in the text.

Thank you for your suggestion, We have added the Fab sequence in the manuscript (Lines 365-372).

10. The angular distribution of particles used in the final reconstructions is missing.

Thank you for your suggestion. We have added the angular distribution of particles used in the final reconstructions to Fig. S3.

11. L. 408 “At the indicated time points” – please specify the time points for each experiment.

Thank you for your suggestion. The actual point of measurement has been included in the manuscript (Line 436).

12. L. 243 – “released” instead of “transported” could be less confusing.

Thank you for your suggestion. We have made the appropriate changes (Line 259).

13. Figure 4d – “Q142N” instead of “N142N”.

Thank you for pointing this out. The typo has been corrected (Fig. 4d).

14. Extended Data Figure 9 – “residue” instead of “resudue”.

Thank you for pointing this out. The typo has been corrected (Fig. S9).

REVIEWERS' COMMENTS

Reviewer #1 (Remarks to the Author):

The manuscript is much improved. All my concerns have been addressed.

Reviewer #2 (Remarks to the Author):

The authors have adequately addressed all of my concerns and with the added analysis this work is of value as the basis for more elaborated investigations of the transport mechanism.

One minor thing - the statement on lines 199-200: 'This structure is virtually identical to our structure, thus

providing support to the observed tetrabenazine binding mode.' sounds logically awkward, better to say that both studies agree well on the binding mode of tetrabenazine.

Reviewer #3 (Remarks to the Author):

The authors responded to all of my comments and have improved the manuscript accordingly.

Reviewer #4 (Remarks to the Author):

The authors fully addressed my concerns and provided explanations to my questions. Upon revision, I find the manuscript to be of excellent quality, presenting a convincing scientific story. I want to congratulate the authors on their work on this important protein.

The authors also improved the proposed transport mechanism, which may contribute to our better understanding of not only the VMAT2 protein but also other members of the SLC and MFS families.